# Utilizing Edge Features in Graph Neural Networks via Variational Information Maximization

## Abstract

Graph Neural Networks (GNNs) broadly follow the scheme that the representation vector of each node is updated recursively using the message from neighbor nodes, where the message of a neighbor is usually pre-processed with a parameterized transform matrix. To make better use of edge features, we propose the Edge Information maximized Graph Neural Network (EIGNN) that maximizes the Mutual Information (MI) between edge features and message passing channels. The MI is reformulated as a differentiable objective via a variational approach. We theoretically show that the newly introduced objective enables the model to preserve edge information, and empirically corroborate the enhanced performance of MI-maximized models across a broad range of learning tasks including regression on molecular graphs and relation prediction in knowledge graphs.

## 1 Introduction

Many real-world datasets naturally come in the form of graphs, such as citation networks (Kipf & Welling, 2017), social networks Hamilton et al. (2017), knowledge graphs (Schlichtkrull et al., 2018), molecular graphs (Scarselli et al., 2009; Duvenaud et al., 2015) etc., all of which consist of a number of nodes and edges equipped with their inherent features. Recently, impressive performance has been achieved in graph learning tasks with various forms of Graph Neural Networks (GNNs) (Zhou et al., 2018). Compared to prior works, such as node2vec (Grover & Leskovec, 2016), GNNs learn the state of a node by recursively aggregating messages from its neighbors: combining the graph structure with node features. Intuitively, edge features should play an important role in graph learning tasks. For example, chemical bonds in a molecule have a high impact on chemical properties of molecules, and edge features in knowledge graphs encode important relations between concepts, data, and entities. Our proposed method focuses on improving the usage of edge features in GNNs.

The expressive power of GNNs largely depends on how the message is passed between nodes. A widely adopted scheme is multiplying neighbor node states with a parameterized transform matrix before aggregation (Gilmer et al., 2017; Xu et al., 2019). Despite tremendous success of GNNs, existing models do not exhaustively exploit the full potentials of edge features on graphs. For example, many GNNs such as GCN (Kipf & Welling, 2017), ChebyNet (Defferrard et al., 2016) and GAT (Veličković et al., 2018) do not even consider categorized edge types. To utilize edge features in multi-relational graphs, RGCN (Schlichtkrull et al., 2018) proposes to learn a different transform matrix for each edge type, respectively. However, it does not generalize to edge features in continuous space. MPNN (Gilmer et al., 2017) introduces an edge network that takes edge feature vectors as input and outputs transform matrices, which are used to transform states of neighbor nodes. In principle, the MPNN framework can handle complex edge features. Yet, the lack of maximization of MI between edge and message channels implies that the MPNN may give an edge-independent transform matrix.

In this work, we aim to more efficiently exploit the full potentials of edge features from the perspective of training. We propose the **E**dge **I**nformation maximized **G**raph **N**eural **N**etwork (EIGNN) that maximizes the Mutual Information (MI) between edge features

and the message passing channel which is parameterized as the transform matrix in the widely-accepted message passing framework (transformation and aggregation) (Gilmer et al., 2017; Xu et al., 2019). Considering the challenge of computing the MI, we adopt a variational approach to reformulate it as an differentiable objective, which can be easily applied as a regularization term. We theoretically show that EIGNN can reduce information loss of edge features. Apart from demonstrating the impressive performance of EIGNN on extensive benchmarks of molecular graphs and knowledge graphs, we also analyze and attribute the enhanced effectiveness of EIGNN to the exploitation of edge features instead of the regularization effects. Notably, attribution analysis on molecular graphs show that EIGNN can capture domain knowledge without human interference.

**Preliminaries** Let $G = (\mathcal{V}, \mathcal{E})$ be a graph with node feature vectors $x_v \in \mathbb{R}^d$ for node $v \in \mathcal{V}$ and edge feature vectors $e_{vw} \in \mathcal{E}$ for the edge connecting node $v$ and $w$. In GNNs, the state of each node is updated recursively using neighbor nodes. Let $\mathcal{N}_v$ be the set of neighbor nodes of $v$ and $h_v^{(l)} \in \mathbb{R}^{d_l}$ be the hidden state of $v$ at $l$-th layer, where $d_l$ is the dimension of the hidden layer. For simplicity of notation, we use a single $d$ to denote the dimension such that $h_v^{(l)} \in \mathbb{R}^d$. We also have $h_v^{(0)} = x_v$ at the input layer.

## 2 Related works

### 2.1 Relational Modeling in Graph Neural Networks

**Single-relational modeling.** Many variants such as GCN (Kipf & Welling, 2017), GAT (Veličković et al., 2018), ChebyNet (Defferrard et al., 2016), GraphSAGE (Hamilton et al., 2017) focus on learning node states. These models can assign weight to neighbors, but they can not handle various edge features. A typical neighborhood aggregation scheme is

$$h_v^{(l+1)} = \sigma \left( \sum\nolimits_{w \in \mathcal{N}_v} \alpha_{vw} W_1^{(l)} h_w^{(l)} + W_0^{(l)} h_v^{(l)} \right), \tag{1}$$

where $\sigma$ denotes an activation function, $\alpha_{vw}$ can be a normalization constant or a learned attention coefficient (Veličković et al., 2018). States of all neighbors are multiplied by the same trainable transform matrix $W_1^{(l)}$. Sometimes the self-connection is also treated in the same way, s.t., $W_0^{(l)} = W_1^{(l)}$.

**Multi-relational modeling.** A simple strategy to handle multi-relational graphs is assigning each edge type with a separate transform matrix as presented in RGCN (Schlichtkrull et al., 2018) and adopted by GGNN (Li et al., 2016) and LNet (Liao et al., 2019). RGCN updates node states according to the following scheme

$$h_v^{(l+1)} = \sigma \left( \sum\nolimits_{r \in \mathcal{R}} \sum\nolimits_{w \in \mathcal{N}_v^r} \alpha_{vw,r} W_r^{(l)} h_w^{(l)} + W_0^{(l)} h_v^{(l)} \right), \tag{2}$$

where $\mathcal{N}_v^r$ is the collection of neighboring nodes of $v$ with relation $r \in \mathcal{R}$ and $\alpha_{vw,r}$ is a normalization constant similar as $\alpha_{vw}$ in Eq. (1). Such a scheme faces challenge in handling edge features of continuous space. GGNN and LNet do not focus on the improvement of edge expressibility. GGNN introduces Gated Recurrent Unit (GRU) (Cho et al., 2014) and LNet focus on handling multi-scale connections.

**Complex-relational modeling.** The relation in a graph can be quite complex, expressed as a general feature vector $e$. MPNN (Gilmer et al., 2017) introduces an edge network which takes edge feature vectors as input and outputs transform matrices. A single edge network is shared in a MPNN model. The forward propagation is formalized as

$$m_v^{(l+1)} = \sigma \left( \sum\nolimits_{w \in \mathcal{N}_v} f(e_{vw}) h_w^{(l)} + W_0^{(l)} h_v^{(l)} \right), \quad h_v^{(l+1)} = \text{GRU}(h_v^{(l)}, m_v^{(l+1)}), \tag{3}$$

where $f : e \to W$ denotes the edge network. Recently, some research works treat a multi-relational problem as the complex-relational one by introducing a continuous edge embedding vector for each edge type, so as to handle increasing number of relations (Nathani et al., 2019). Although the MPNN architecture allows the usage of arbitrary edge features, this advantage is not utilized in practice. MPNN can actually learn an edge-independent transform matrix. A GNN model that efficiently utilizes edge features is yet to emerge.

## 2.2 Readout functions

After several forward propagations, GNNs yield final states of all nodes, which are suitable for node/edge classification or regression. For graph classification or regression tasks, we can apply a readout function (Ying et al., 2018; Vinyals et al., 2015) such that

$$y = R(\{h_v^L | v \in G\}),  \tag{4}$$

where $h_v^L$ is the state of $v$ at the last layer, $R$ is the readout function that outputs a graph-level representation $y$, e.g., summing up the final node states, applying hierarchical pooling (Ying et al., 2018) or using the set2set model (Vinyals et al., 2015).

## 3 Our method

### 3.1 The Usage of Mutual Information

In probability theory and information theory, MI is a measure of mutual dependence between two random variables. Our method proposes to preserve edge information in GNNs, which is important in many real-world graph structures such as molecules - apart from node (atom) features, attributes of edges (bonds) are equally important for predicting properties of molecules. To this end, we maximize $I(e; W)$ - the MI between the edge feature vector $e$ and the message passing channel, i.e., the transform matrix $W$ which is used to transform neighbor node states in the forward propagation. Our method can be easily generalized to directly maximize the MI between edge features and the message itself in methods that do not explicitly have the transform matrix, e.g., the message from node $w$ to node $v$ can be expressed as $f(h_v, e_{vw}, h_w)$ rather than $f(e_{vw})h_w$, which is shown in Section 4.3.

An general principle of maximum MI is described for unsupervised learning task by (Linsker, 1988) and MI inspired objective functions have long been adopted in unsupervised learning (Bridle et al., 1992; Barber & Agakov, 2006; Veličković et al., 2019; Hjelm et al., 2019), semi-supervised learning (Krause et al., 2010) and generative adversarial networks (Chen et al., 2016). Specifically, DGI (Veličković et al., 2019) also applies MI to GNNs. DGI proposes to learn node-wise representations in an unsupervised manner by maximizing the MI between node representations and corresponding high-level summaries of graphs, using adversarial learning and negative sampling. The node representations may then be retrieved and used for downstream tasks, such as node classification. DGI can be used to pre-train GNNs, as demonstrated in Hu et al. (2019). Our EIGNN also proposes information maximization but targets a completely different objective and adopts a quite different approach.

### 3.2 A Variational Approach to Maximize Mutual Information

Computing $I(e; W)$ itself is intractable in practice, needless to say that training a model requires the derivative. Thus, we adopt a variational approach (Agakov, 2004) to reformulate $I(e; W)$ as a differentiable objective. We show that our objective is an approximated lower bound of $I(e; W)$ and notably, optimizing our objective does lead to maximizing $I(e; W)$. Following MPNN (Gilmer et al., 2017), we use an edge network to parameterize the transform matrix $W$ and relate it to edge features. Therefore, the prior $p(W|e)$ is

$$p(W|e) = \delta(W - f(e)),  \tag{5}$$

where $\delta(\cdot)$ is the Dirac delta function. The posterior $p(e|W)$ is intractable, so we define a variational distribution $q(e|W)$, which can be obtained by defining a neural network $g : W \to e$. Specifically, $q(e|W)$ substitutes to some distribution (such as Gaussian distribution) with parameter $g(W)$. In this way, $f$ and $g$ are similar to the probabilistic encoder and decoder in the Variational Auto-Encoder (VAE) (Kingma & Welling, 2013). Then we can approximate $I(e; W)$ with a differentiable objective $L_I(f, g; e)$ as follows.

**Theorem 1.** *Let $e$ be the edge feature vector, $W$ be the transform matrix with conditional distribution $p(W|e)$ specified by the probabilistic encoder $f$ as shown in Eq. (5) and $q(e|W)$ be the variational distribution specified by the probabilistic decoder $g$, then we have*

$$I(e; W) \geq H(e) + \mathbb{E}_{e \sim p(e)}[\mathcal{L}_I(f, g; e)],  \tag{6}$$

*where $\mathcal{L}_I(f, g; e) = \log q(e|f(e))$ and $H(\cdot)$ denotes the entropy.*

*Proof.* Let $D_{KL}(\cdot \parallel \cdot)$ denote the KL-divergence, which should be nonnegative, then we have

$$
\begin{aligned}
I(e; W) &= H(e) - H(e|W) \\
&= H(e) + \mathbb{E}_{W \sim p(W)}[\mathbb{E}_{e \sim p(e|W)}[\log p(e|W)]] \\
&= H(e) + \mathbb{E}_{W \sim p(W)}[\mathbb{E}_{e \sim p(e|W)}[\log p(e|W) - \log q(e|W) + \log q(e|W)]] \\
&= H(e) + \mathbb{E}_{W \sim p(W)}[D_{KL}(p(e|W) \parallel q(e|W)) + \mathbb{E}_{e \sim p(e|W)}[\log q(e|W)]] \\
&\geq H(e) + \mathbb{E}_{W \sim p(W)}[\mathbb{E}_{e \sim p(e|W)}[\log q(e|W)]] \\
&= H(e) + \mathbb{E}_{e \sim p(e), W \sim p(W|e)}[\log q(e|W)] \\
&\overset{(a)}{=} H(e) + \mathbb{E}_{e \sim p(e)}[\log q(e|f(e))]
\end{aligned}
$$

where the equality $(a)$ follows from Eq. (5). $\qquad\square$

According to Theorem 1, we can maximize the variational lower bound for $I(e; W)$. The bound becomes tight when the variational distribution $q(e|W)$ approaches the true posterior $p(e|W)$. Moreover, $H(e)$ is a constant because the distribution of edge feature vector $e$ is fixed for given graphs, hence we can equivalently maximize $\mathcal{L}_I(f, g; e)$. We choose the widely accepted Gaussian distribution as the prior distribution for the probabilistic decoder $g$,

$$
q(e|W) = \mathcal{N}(e; g(W), \sigma^2 I). \tag{7}
$$

Then we have

$$
\mathcal{L}_I(f, g; e) = \log q(e|f(e)) = \log \mathcal{N}(e; g(f(e)), \sigma^2 I) = -\lambda \|e - g(f(e))\|_2^2 \tag{8}
$$

where $\lambda > 0$ is a constant determined by $\sigma$ and the dimension of $e$, taken as a tunable parameter. The following Theorem 2 shows that maximizing the objective in Eq. (8) does lead to the maximization of $I(e; W)$, hence enables the model to preserve edge information.

**Theorem 2.** *Assume the optimal solution of maximizing $\mathcal{L}_I(f, g; e)$ is $f^\star$ and $g^\star$, then $f^\star$ also maximizes $I(e; W)$.*

*Proof.* Note that $H(e)$ is a constant when the graphs are given. In information theory, we have

$$
H(g(f(e))) \leq H(f(e)) \leq H(e). \tag{9}
$$

$I(e; W)$ is upper bounded by $H(e)$,

$$
I(e; W) = I(e; f(e)) = H(f(e)) - H(f(e)|e) = H(f(e)) \leq H(e).
$$

Since $f^\star$ and $g^\star$ is the optimal solution of maximizing $\mathcal{L}_I(f, g; e)$ presented in Eq. (8), it is not difficult to see that $e = g^\star(f^\star(e)), \forall e \in \mathcal{E}$. In this case, the inequalities in Eq. 9 become equalities, i.e.,

$$
H(g^\star(f^\star(e))) = H(f^\star(e)) = H(e).
$$

Therefore, we have $I(e; W) = H(e)$, i.e., the maximum is attained in this case. $\qquad\square$

### 3.3 Edge Information Maximized Graph Neural Networks

Our EIGNN is derived by implementing our MI objective in GNNs. As a concrete example, the forward propagation of our model follows the formulation in Eq. (3), where the dege network $f : e \to W$ is expressed as a multi-layer perceptron (MLP). According to theoretical analysis presented in Sec. 3.2 , we introduce another MLP $g : W \to e$ as the decoder.

For graph regression or classification tasks, the model outputs a prediction $y$ for each graph $G$, which has label $\hat{y}$. Without MI maximization, we denote the vanilla loss as $\mathcal{L}_0(\hat{y}, y; G)$. Common choice of $\mathcal{L}_0$ includes Mean Square Error (MSE), Mean Absolute Error (MAE) and Cross Entropy (CE). For a graph $G = (\mathcal{V}, \mathcal{E})$, EIGNN maximizes $\mathcal{L}_I(f, g; e)$ and minimizes $\mathcal{L}_0(\hat{y}, y; G)$ using the following loss function

$$
\mathcal{L}_{EIGNN}(G) = \mathcal{L}_0(\hat{y}, y; G) - \lambda \mathbb{E}_{e \in \mathcal{E}}[\mathcal{L}_I(f, g; e)], \tag{10}
$$

where $\mathbb{E}_{e \in \mathcal{E}}[\cdot]$ denotes taking the mean over all edges in $G = (\mathcal{V}, \mathcal{E})$ and $\lambda$ is the regularization parameter. When EIGNN is trained using mini-batches, $\mathcal{L}_0(\hat{y}, y; G)$ is averaged over all graphs in the batch while $\mathcal{L}_I(f, g; e)$ is averaged over all edges of all graphs in the batch.

Similarly, for relational prediction tasks in knowledge graphs, EIGNN directly yields node-level representations $h_v$ for each node $v \in \mathcal{V}$ and edge-level representations $e$ for each relationship. The objective function of EIGNN can be derived from the translational scoring function Bordes et al. (2013), which learns embedding such that for a given valid triple $t_{vw} = (h_v, e_{vw}, h_w)$ from the valid set $S$, the condition $d_{t_{vw}} = h_v + e_{vw} - h_w \approx 0$ holds. Let $\mathcal{L}_0 = \mathbb{E}_{t_{vw} \in S} \mathbb{E}_{t'_{vw} \in S'} \max\{d_{t'_{vw}} - d_{t_{vw}} + \gamma, 0\}$, where $S'$ denotes a set of invalid triples and $\gamma$ is a margin hyper-parameter, EIGNN can be trained by minimizing the following loss,

$$\mathcal{L}_{EIGNN}(G) = \mathcal{L}_0 - \lambda \mathbb{E}_{e \in \mathcal{E}} [\mathcal{L}_I(f, g; e)]. \tag{11}$$

## 4 EXPERIMENTS

In this section, we first conduct experiments on a large quantum chemistry benchmark QM9, which is challenging for most baselines. Then we evaluate EIGNN on several useful molecule benchmarks and use attribution analysis to show that EIGNN increases the impact of edges and captures domain knowledge without human interference. Finally, we adopt our method to large-scale knowledge graphs and evaluate the performance on challenging relation prediction tasks using a wide variety of real-world datasets. All experimental results demonstrate a clear and substantial improvement of EIGNN over the state-of-the-art methods.

### 4.1 QUANTUM CHEMISTRY

QM9 (Ramakrishnan et al., 2014) is a large benchmark containing 134k molecules with 12 quantum chemistry regression properties, which have been show to be quite challenging for many GNNs (Gilmer et al., 2017). Feature engineering of nodes and edges exactly follows (Gilmer et al., 2017) such that molecules are preprocessed as graphs according to atom features and bond features. We compare our EIGNN with nine state-of-the-art baselines which can be categorized into three groups according to the ability of handling edge features: **i)** GCN, ChebyNet, GAT and GIN (Xu et al., 2019) which simply use binary edge features to indicate the existence of a bond without any other edge features; **ii)** RGCN, GGNN, LNet and simplified MPNN (sMPNN) which consider bond types (no bond, single, double, triple, or aromatic); **iii)** MPNN and our EIGNN which use edge feature vectors to indicate both edge types and pairwise distance between atoms.

For a fair comparison, we repeat all experiments 3 times with different random seeds while during each run, all methods share the same random seed. We randomly choose 10k molecules for validation, 10k molecules for testing, and keep the rest for training. Each target property is normalized to zero mean and unit variance for training. Each model is trained to predict the 12 target properties simultaneously. $\lambda$ is naively set to 1 for EIGNN. We use mean square error (MSE) loss to train the models for at most 300 epochs till convergence, and the performance is measured by mean absolute error (MAE). For LNet and GGNN, implementation of the readout function follows the original paper. While for all other models,

Table 1: Quantum property regressions for 12 targets and overall performance (top two raws) on QM9. We repeat all experiments 3 times with different random seeds and report the average performance. Full results with standard deviation are presented in Appendix A, e.g., for MPNN and EIGNN, we have Avg. nMAE $0.0398 \pm 0.0002$ and $0.0357 \pm 0.0005$.

| Method | GCN | ChebyNet | GAT | GIN | RGCN | GGNN | LNet | sMPNN | MPNN | EIGNN |
|---|---|---|---|---|---|---|---|---|---|---|
| Avg. nMAE | 0.135 | 0.121 | 0.137 | 0.100 | 0.102 | 0.099 | 0.099 | 0.089 | 0.040 | **0.036** |
| Avg. MAE | 5.306 | 4.303 | 5.470 | 3.480 | 3.817 | 3.661 | 3.653 | 3.161 | 0.693 | **0.633** |
| mu | 0.568 | 0.518 | 0.567 | 0.478 | 0.506 | 0.518 | 0.472 | 0.472 | 0.110 | **0.097** |
| alpha | 0.881 | 0.793 | 0.891 | 0.621 | 0.632 | 0.608 | 0.623 | 0.528 | 0.332 | **0.294** |
| HOMO($10^{-3}$) | 5.451 | 4.775 | 5.429 | 4.183 | 4.453 | 4.483 | 3.889 | 3.854 | 2.481 | **2.230** |
| LUMO($10^{-3}$) | 6.400 | 5.674 | 6.331 | 4.796 | 5.138 | 5.153 | 4.194 | 4.549 | 2.862 | **2.593** |
| gap($10^{-3}$) | 8.201 | 7.097 | 8.193 | 6.096 | 6.500 | 6.602 | 5.813 | 5.634 | 3.620 | **3.275** |
| R2 | 53.56 | 41.95 | 54.52 | 34.65 | 40.10 | 39.68 | 35.27 | 33.49 | 6.064 | **5.646** |
| ZPVE($10^{-3}$) | 2.533 | 2.527 | 2.271 | 1.744 | 1.477 | 1.292 | 1.438 | 1.345 | 0.679 | **0.612** |
| U0 | 2.042 | 1.984 | 2.290 | 1.422 | 1.059 | 0.697 | 1.806 | 0.791 | 0.416 | **0.357** |
| U | 2.042 | 1.984 | 2.290 | 1.422 | 1.059 | 0.697 | 1.755 | 0.791 | 0.416 | **0.357** |
| H | 2.042 | 1.984 | 2.290 | 1.422 | 1.059 | 0.697 | 1.796 | 0.791 | 0.416 | **0.357** |
| G | 2.042 | 1.984 | 2.290 | 1.422 | 1.059 | 0.696 | 1.778 | 0.791 | 0.416 | **0.357** |
| Cv | 0.473 | 0.420 | 0.479 | 0.309 | 0.317 | 0.315 | 0.312 | 0.262 | 0.134 | **0.121** |

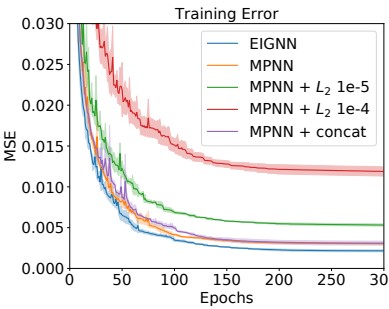 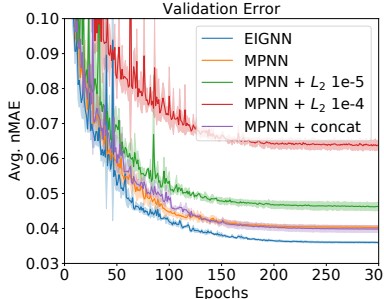

Figure 1: Ablation study. Training and validation error on QM9. The shadow area indicates $mean \pm std$ over 3 runs.

we use the same set2set (Vinyals et al., 2015) readout, which has been demonstrated to work well in (Gilmer et al., 2017).

In Table 1, we list regression results for all methods. We report individual MAE for each target in their original scale, averaged MAE (Avg. MAE) over 12 properties, and averaged normalized MAE (Avg. nMAE; averaged over normalized target properties since different targets have different units and ranges). Our EIGNN achieves the best performance for each metric and each target. Now we are ready to answer the following research questions. **i) Are edge features important?** Yes. The error has a trend of decreasing with increasing edge features. The comparison between sMPNN (using edge types) and MPNN (using edge types and distance) directly verifies the importance of edge features. It is also consistent with the expert knowledge that distances between pairwise atoms are closely related to quantum properties. For example, the smaller the distance between the two atoms, the stronger the bond is, and consequently a higher bond energy is associated with this atom pair. **ii) Does the EIGNN work?** Yes. EIGNN achieves the best performance on each target, outperforming the strong baseline MPNN. Moreover, the advantage of EIGNN over MPNN is consistent over 3 runs and the standard deviation on this task is quite small. Detailed results are shown in Table 4 of Appendix A. **iii) How does the EIGNN work?** Our MI objective is easily implemented on top of vanilla loss function. We have shown that our objective enables preserving of edge information. Fig. 1 demonstrates that regularization such as $L_2$ weight decay can increase training error while our objective does not. Moreover, the validation performance verifies that regularization itself does not reduce the validation error. Thus, the effectiveness of EIGNN is due to exploiting edge features rather than the regularization effect. We further run an ablation study where we concatenate edge features to node representations (i.e., MPNN+concat in Fig. 1) in message passing. Concatenation is unable to identify correlations between edges and nodes (Gilmer et al., 2017) and our results show that it slightly reduces the mean validation error but increases the variance.

## 4.2 More Molecule Benchmarks with Potential Applications

We further evaluate EIGNN on three molecule benchmarks: Lipophilicity (Wu et al., 2018), ESOL (Delaney, 2004) and FreeSolv (Mobley & Guthrie, 2014). These datasets contain fewer molecules, and have potential usages in applications such as chemistry, drug discovery, and materials science. For example, the property lipophilicity is an important feature of drug molecules that affects both membrane permeability and solubility. The dataset Lipophilicity contains 4200 compounds. ESOL provides water solubility data for 1128 compounds. FreeSolv contains hydration free energy of 642 small molecules in water. We conduct graph regression experiments on these benchmarks. All datasets are split into training, validation and test according to a proportion of 0.8/0.1/0.1. MPNN and our EIGNN share the same architecture with 3 layers of message passing and 3 steps of set2set. We repeat each experiment 3 times

Table 2: Testing RMSE on Lipophilicity, ESOL and FreeSolv.

| Dataset | Lipophilicity | | ESOL | | FreeSolv | |
|---|---|---|---|---|---|---|
| Method | MPNN | EIGNN | MPNN | EIGNN | MPNN | EIGNN |
| mean±std | 0.678±0.042 | **0.653±0.025** | 0.805±0.064 | **0.776±0.071** | 1.398±0.081 | **1.273±0.137** |

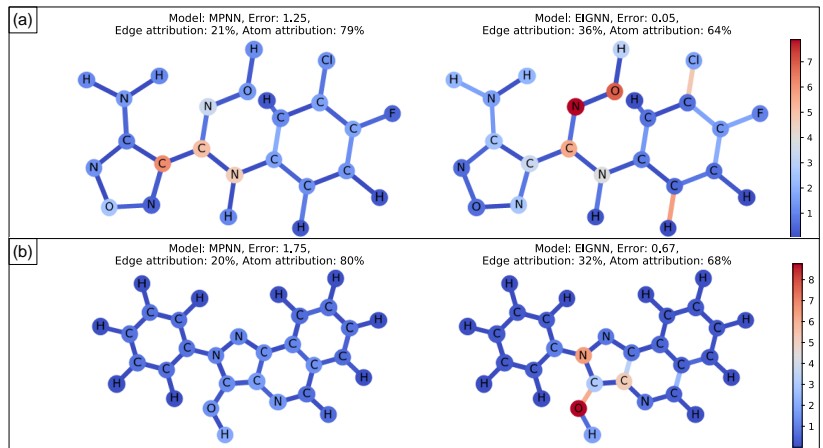

Figure 2: Attribution analysis. The color indicates the impact of an edge/atom on the output, i.e., the regression result. EIGNN i) increases the edge attribution, ii) reduces the prediction error and iii) can learn domain knowledge without human interference.

with different random seeds. Results of testing root mean square error (RMSE) in Table 2 verify the effectiveness of our method. Our EIGNN outperforms MPNN on each dataset and each run. Detailed results for each run are presented in Appendix B.

**Attribution analysis.** To understand how our EIGNN reduces the regression error, we conduct attribution analysis, i.e., attributing the prediction of a deep network to its input features, which usually builds up on the standard gradient operator (Sundararajan et al., 2017). For an output $y$, i.e., the prediction of a GNN, we define its sensitivity to an edge with features $e$ as $S_e = |\frac{\partial y}{\partial e}|$, where $|\cdot|$ denotes the $L_1$ norm. Similarly, the sensitivity to an atom with features $x$ is $S_x = |\frac{\partial y}{\partial x}|$. Then $S_e$ and $S_x$ are used as the metrics of attribution in our experiments. As an example, we show in Fig. 2 the attribution for two molecules in Lipophilicity: (a) *Nc1nonc1C(=NO)Nc2ccc(F)c(Cl)c2* and (b) *Oc1c2ncc3ccccc3c2nn1c4ccccc4* . Molecule in (a) has the potential to be used to treat, prevent and/or diagnose cancer Prinz et al. (2019). Compared with MPNN, we can observe an increasing of overall edge attribution under our EIGNN and a decreasing of prediction error in both cases. Interestingly, the attribution under EIGNN is similar to the expert knowledge of chemists: halogen atoms such as {Cl, Br, I} and their bond with the carbon atom greatly effect the lipophilicity of a molecule Wilcken et al. (2013), while atoms {O, N} also have high impact on the lipophilicity but usually in a negative way (Augustijns & Brewster, 2007). In Fig. 2 (a), the attribution of the halogen bond C-Cl and the pair {O, N} under our EIGNN is much higher than the one under MPNN, which is consistent with the expert knowledge. In Fig. 2 (b), similarly, the attribution of atoms {O, N} and the bond C-O under EIGNN is much higher. More examples are presented in Appendix C.

### 4.3 Predicting Relations in Knowledge Graphs

In this subsection, we adopt EIGNN to tackle the problem of relation prediction in knowledge graphs (KGs), which entails predicting whether a given triple is valid or not. For example, a triple *(London, capital of, United Kingdom)* should be classified as valid or *London* should be predicted as the *capital of United Kingdom*. KGs represent human knowledge as a directed graph, and have been widely used in practical applications, such as semantic search, dialogue generation, question answering etc. Recovering missing relations in KGs have been a major task for practical usages of KGs. We evaluate our methods on three benchmark datasets, WN18RR (Dettmers et al., 2018), FB15k-237 (Toutanova et al., 2015) and NELL-995 (Xiong et al., 2017). Without the reversible relation problem (Dettmers et al., 2018), WN18RR includes 11 relations scraped from WordNet for $40,943$ synsets. FB15k-237 is a subset of Freebase, and contains $14,541$ entities associated with $237$ types of edge. NELL-995 is constructed from the $995^{th}$ iteration of NELL system, containing $75,492$ entities and $200$ types of edge.

Table 3: Experimental results on WN18RR, FB15K-237 and NELL-995 test sets. Hits@N values are in percentage. The best score is in bold and second best score is underlined.

| Dataset | WN18RR | | | | FB15K-237 | | | | NELL-995 | | | |
|---|---|---|---|---|---|---|---|---|---|---|---|---|
| | | Hit@N % | | | | Hit@N % | | | | Hit@N % | | |
| | MRR | @1 | @3 | @10 | MRR | @1 | @3 | @10 | MRR | @1 | @3 | @10 |
| DistMult | 0.444 | 41.2 | 47 | 50.4 | 0.281 | 19.9 | 30.1 | 44.6 | 0.485 | 40.1 | 52.4 | 61 |
| ComplEx | 0.449 | 40.9 | 46.9 | 53 | 0.278 | 19.4 | 29.7 | 45 | 0.482 | 39.9 | 52.8 | 60.6 |
| ConvE | **0.456** | 41.9 | 47 | 53.1 | 0.312 | 22.5 | 34.1 | 49.7 | 0.491 | 40.3 | 53.1 | 61.3 |
| TransE | 0.243 | **42.7** | 44.1 | 53.2 | 0.279 | 19.8 | 37.6 | 44.1 | 0.401 | 34.4 | 47.2 | 50.1 |
| ConvKB | 0.265 | 58.2 | 44.5 | 55.8 | 0.289 | 19.8 | 32.4 | 47.1 | 0.43 | 37.0 | 47 | 54.5 |
| R-GCN | 0.123 | 8 | 13.7 | 20.7 | 0.164 | 10 | 18.1 | 30 | 0.12 | 8.2 | 12.6 | 18.8 |
| KBGAT | 0.436 | 35.8 | 48.1 | 57.8 | 0.431 | 36.1 | 45.8 | 56.9 | 0.514 | 42.9 | 55.3 | 67.8 |
| EIGNN | 0.438 | 35.7 | **48.8** | **58.1** | **0.451** | **37.4** | **48.2** | **60.5** | **0.523** | **43.8** | **56.1** | **68.3** |

A critical issue of applying Eq. (3) to KGs is that high-dimensional embedding vectors are required to distinguish massive amount of entities and relations, leading to a rapid growth in number of parameters in EIGNN. To address this issue, we adopt the architecture of (Nathani et al., 2019) and learn graph attention based embeddings that target relation prediction on KGs as follows,

$$m_{vw} = f(h_v^{(l)}, e_{vw}^l, h_w^{(l)}), \; \alpha_{vw}^l = \mathrm{softmax}(a^l m_{vw}^l), \; h_v^{(l+1)} = \sigma\left(\sum_{w \in \mathcal{N}_v} \alpha_{vw}^l m_{vw}^l\right). \quad (12)$$

Compared with Eq. (3), where $m_{vw} = f(e_{vw})h_w$, the above equation absorbs the transform matrix $f(e_{vw})$ into $f(h_v^{(l)}, e_{vw}, h_w^{(l)})$ and reduces the model parameters. In the following experiments, we implement $f$ using a MLP as in previous experiments, and maximizes the MI between $e_{vw}$ and $m_{vw}$ by introducing another MLP with $\lambda = 0.01$. Multi-head attention is further introduced to stabilize the learning process and encapsulate more information about neighbors according to (Veličković et al., 2018). After training EIGNN, ConvKB (Nguyen et al., 2018) is adopted as a regression function for a given triple by analyzing the global embedding properties across each dimension.

In the relation prediction task, the aim is to predict a triple $(v, e_{vw}, w)$ with $v$ or $w$ missing. We can generate a set of candidate triples for each missing entity $v$ by randomly replacing it with an arbitrary one. Scores can be calculated by ConvKB for all triples, and we find the rank of a correct triple by sorting all scores in ascending order. Thus, the performance of relation prediction task can be evaluated by mean reciprocal rank (MRR) and the proportion of correct entities in the top $N$ ranks (Hits@N) for $N = 1, 3$, and 10 (Bordes et al., 2013). We compare our EIGNN with seven state-of-the-art baselines focusing on this task: DistMult (Yang et al., 2014), ComplEx (Trouillon et al., 2016), ConvE (Dettmers et al., 2018), TransE (Bordes et al., 2013), ConvKB (Nguyen et al., 2018), RGCN (Schlichtkrull et al., 2018) and KBGAT (Nathani et al., 2019). As shown in Table 3, our EIGNN achieves the best performance for each metric on FB15K-237 and NELL-995, and achieves the best performance on WN18RR with Hit@3 and 10 metrics. The results of KBGAT are reproduced following the official implementation[1], and the results of other methods can be found in the previous peer-reviewed publications, i.e. (Nathani et al., 2019).

## 5 Conclusions

In this work, to make better use of edge features in GNNs, we proposed the edge information maximized graph neural network (EIGNN) that maximizes the mutual information between edge feature vectors and message passing channels. We reformulated the mutual information as a differentiable objective by adopting a variational approach. We have theoretically proved that our proposed objective enables EIGNN to preserve edge information and empirically evaluated EIGNN's performance on a variety of benchmarks incorporating an array of challenging molecular datasets and knowledge graphs. These results clearly manifested a substantial improvement of EIGNN over the prior state-of-the-art methods. Apart from demonstrating the impressive performance of EIGNN, we also showed that its effectiveness is due to exploitation of edge features instead of the regularization effect. Notably, attribution analysis on molecular graphs show that EIGNN can capture domain knowledge in an end-to-end fashion.

---

[1]https://github.com/deepakn97/relationPrediction

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

## A  Full Results on QM9

Table 4: Full results of quantum property regressions for 12 targets and overall performance (nMAE and MAE in top two raws) on QM9. We repeat all experiments 3 times with different random seeds and report the average performance and standard deviation. This is a supplement for Table 1. in the main text.

| Method | GCN | ChebyNet | GAT | GIN |
|---|---|---|---|---|
| Avg. nMAE | 0.1350±0.0046 | 0.1206±0.0084 | 0.1367±0.0050 | 0.1001±0.0007 |
| Avg. MAE | 5.3063±0.1964 | 4.3032±0.4814 | 5.4698±0.2040 | 3.4799±0.0402 |
| mu | 0.5679±0.0078 | 0.5180±0.0131 | 0.5670±0.0102 | 0.4783±0.0041 |
| alpha | 0.8811±0.0308 | 0.7932±0.0778 | 0.8913±0.0314 | 0.6209±0.0028 |
| HOMO($10^{-3}$) | 5.4510±0.0790 | 4.7750±0.2040 | 5.4290±0.1660 | 4.1830±0.0370 |
| LUMO($10^{-3}$) | 6.4000±0.1230 | 5.6740±0.2670 | 6.3310±0.2390 | 4.7960±0.0520 |
| gap($10^{-3}$) | 8.2010±0.2150 | 7.0970±0.3620 | 8.1930±0.2910 | 6.0960±0.0420 |
| R2 | 53.563±1.0319 | 41.950±4.8289 | 54.519±1.5992 | 34.647±0.2167 |
| ZPVE($10^{-3}$) | 2.5330±0.1070 | 2.5270±0.3560 | 2.2710±0.1450 | 1.7440±0.0100 |
| U0 | 2.0422±0.3281 | 1.9842±0.2035 | 2.2899±0.2000 | 1.4215±0.0857 |
| U | 2.0422±0.3281 | 1.9842±0.2035 | 2.2899±0.2000 | 1.4215±0.0857 |
| H | 2.0422±0.3281 | 1.9842±0.2035 | 2.2899±0.2000 | 1.4215±0.0857 |
| G | 2.0423±0.3281 | 1.9842±0.2036 | 2.2899±0.2000 | 1.4215±0.0857 |
| Cv | 0.4730±0.0209 | 0.4199±0.0499 | 0.4787±0.0164 | 0.3093±0.0035 |

| Method | RGCN | GGNN | LNet | sMPNN |
|---|---|---|---|---|
| Avg. nMAE | 0.1021±0.0016 | 0.0992±0.0013 | 0.0992±0.0061 | 0.0888±0.0014 |
| Avg. MAE | 3.8175±0.0605 | 3.6608±0.0723 | 3.6527±0.3417 | 3.1610±0.0697 |
| mu | 0.5056±0.0048 | 0.5179±0.0076 | 0.4717±0.0063 | 0.4718±0.0096 |
| alpha | 0.6321±0.0145 | 0.6077±0.0092 | 0.6225±0.0508 | 0.5278±0.0106 |
| HOMO($10^{-3}$) | 4.4530±0.1290 | 4.4830±0.0650 | 3.8889±0.1617 | 3.8540±0.0410 |
| LUMO($10^{-3}$) | 5.1380±0.1210 | 5.1530±0.0890 | 4.1935±0.2205 | 4.5490±0.0810 |
| gap($10^{-3}$) | 6.5000±0.1390 | 6.6020±0.1400 | 5.8132±0.6456 | 5.6340±0.0570 |
| R2 | 40.102±0.7428 | 39.685±0.8212 | 35.275±3.0531 | 33.489±0.6562 |
| ZPVE($10^{-3}$) | 1.4770±0.0090 | 1.2920±0.0340 | 1.4376±0.0769 | 1.3450±0.0260 |
| U0 | 1.0589±0.0231 | 0.6969±0.0413 | 1.8058±0.2533 | 0.7914±0.0446 |
| U | 1.0589±0.0231 | 0.6966±0.0418 | 1.7555±0.2196 | 0.7914±0.0446 |
| H | 1.0589±0.0231 | 0.6974±0.0408 | 1.7964±0.2428 | 0.7914±0.0446 |
| G | 1.0589±0.0231 | 0.6961±0.0421 | 1.7780±0.2458 | 0.7914±0.0446 |
| Cv | 0.3170±0.0152 | 0.3146±0.0125 | 0.3124±0.0303 | 0.2625±0.0040 |

| Method | MPNN | EIGNN |
|---|---|---|
| Avg. nMAE | 0.0398±0.0002 | 0.0357±0.0005 |
| Avg. MAE | 0.6929±0.0212 | 0.6331±0.0298 |
| mu | 0.1095±0.0014 | 0.0974±0.0026 |
| alpha | 0.3318±0.0026 | 0.2939±0.0054 |
| HOMO($10^{-3}$) | 2.4810±0.0200 | 2.2300±0.0310 |
| LUMO($10^{-3}$) | 2.8620±0.0370 | 2.5930±0.0440 |
| gap($10^{-3}$) | 3.6200±0.0180 | 3.2750±0.0520 |
| R2 | 6.0637±0.2511 | 5.6464±0.3098 |
| ZPVE($10^{-3}$) | 0.6790±0.0140 | 0.6120±0.0170 |
| U0 | 0.4164±0.0225 | 0.3574±0.0100 |
| U | 0.4164±0.0225 | 0.3575±0.0100 |
| H | 0.4164±0.0225 | 0.3574±0.0100 |
| G | 0.4164±0.0225 | 0.3575±0.0101 |
| Cv | 0.1339±0.0013 | 0.1208±0.0027 |

We present full results of quantum property regressions on QM9 with average performance and standard deviation in Table 4. The ($10^{-3}$) in the parentheses indicates that the values in the corresponding raw of the table should multiply by $10^{-3}$. This is simply for clear presentation of the values. We also present a detailed descriptions on the target properties in Table 5 for your reference.

In our results, we directly report MAE instead of Error Ratio [(MAE)/(Chemical Accuracy) (Gilmer et al., 2017)], because it is common to report MAE in terms of chemical unit. This practice has been widely adopted not only in computational chemistry but also in the machine learning community working on molecular graphs (Wu et al., 2018; Morris et al., 2019). Still, we add Table 6 which contains the Error Ratio for MPNN and our EIGNN. In this comparison, we follow (Gilmer et al., 2017) and train models separately to predict each target. Our EIGNN consistently outperforms MPNN.

Table 5: Regression targets on QM9.

| Target property | Description | Unit |
|---|---|---|
| mu | Dipole moment | D |
| alpha | Isotropic polarizability | $a_0^3$ |
| HOMO | Highest occupied molecular orbital energy | $E_h$ |
| LUMO | Lowest unoccupied molecular orbital energy | $E_h$ |
| gap | Gap between HOMO and LUMO | $E_h$ |
| R2 | Electronic spatial extent | $a_0^2$ |
| ZPVE | Zero point vibrational energy | $E_h$ |
| U0 | Internal energy at 0K | $E_h$ |
| U | Internal energy at 298.15K | $E_h$ |
| H | Enthalpy at 298.15K | $E_h$ |
| G | Free energy at 298.15K | $E_h$ |
| Cv | Heat capavity at 298.15K | $\frac{cal}{molK}$ |

Table 6: Error Ratio [(MAE)/(Chemical Accuracy) (Gilmer et al., 2017)] on QM9. Note that the energy values of {U0, U, H, G} are per molecule rather than per atom. Following (Gilmer et al., 2017), models are separately trained on each target.

| Target | MPNN | EIGNN | Improvement (%) |
|---|---|---|---|
| mu | 0.87 | **0.80** | 8.24 |
| alpha | 2.64 | **2.44** | 7.57 |
| HOMO | 1.54 | **1.39** | 10.2 |
| LUMO | 1.31 | **1.28** | 2.13 |
| gap | 2.20 | **1.97** | 10.4 |
| R2 | 1.00 | **0.72** | 28.4 |
| ZPVE | 5.31 | **4.43** | 16.5 |
| U0 | 64.3 | **35.8** | 44.4 |
| U | 56.6 | **23.6** | 58.3 |
| H | 77.4 | **35.2** | 54.5 |
| G | 46.6 | **34.4** | 26.1 |
| Cv | **1.69** | 1.79 | -6.22 |

# B    Full Results on Lipophilicity, ESOL and FreeSolv

In this section, we present experimental results on Lipophilicity, ESOL and FreeSolv with detailed results for each run. The results in Table 7 verify that our EIGNN consistently outperforms MPNN.

# C    More Examples of Attribution

In this section, we present more examples of attribution analysis on Lipophilicity. As a supplement for Fig. 2 in the main text, the observation here is similar. Compared with MPNN, we can observe an increasing of overall edge attribution under our EIGNN and a decreasing of prediction error in both cases. Notably, our EIGNN is able to capture the expert knowledge. **(a)** The molecule is $CN[C@@H](C)C(=O)N[C@@H](C1CCCCC1)C(=O)N[C@H]2CCCN(CCc3ccc(F)cc3)C2$. The attribution of {O, N} and the halogen atom F is higher under EIGNN. **(b)** The molecule is $CC(C)N1CCN[C@H](C1)C(=O)N2CCN(CC2)C(=O)Nc3ccc(Cl)c(Cl)c3$. Our EIGNN successfully captures the importance of two critical halogen atoms Cl and several atoms N. **(c)** The molecule is $COc1ccc(cc1)C(=O)N2CCCC2 = O$. The attribution of two atoms O at the top is

Table 7: Testing RMSE on Lipophilicity, ESOL and FreeSolv. This is a supplement for Table 2 in the main text.

| Dataset | Lipophilicity | | ESOL | | FreeSolv | |
|---|---|---|---|---|---|---|
| Seed | MPNN | EIGNN | MPNN | EIGNN | MPNN | EIGNN |
| 0 | 0.718 | **0.676** | 0.770 | **0.718** | 1.396 | **1.109** |
| 1 | 0.696 | **0.664** | 0.750 | **0.733** | 1.299 | **1.265** |
| 2 | 0.620 | **0.619** | 0.894 | **0.876** | 1.499 | **1.443** |
| mean±std | 0.678±0.042 | **0.653±0.025** | 0.805±0.064 | **0.776±0.071** | 1.398±0.081 | **1.273±0.137** |

much higher under EIGNN. **(d)** The molecule is $Cc1cc2NC(=O)C(=CC(=O)c2cc1C)O$. The attribution of atoms O is much higher under EIGNN.

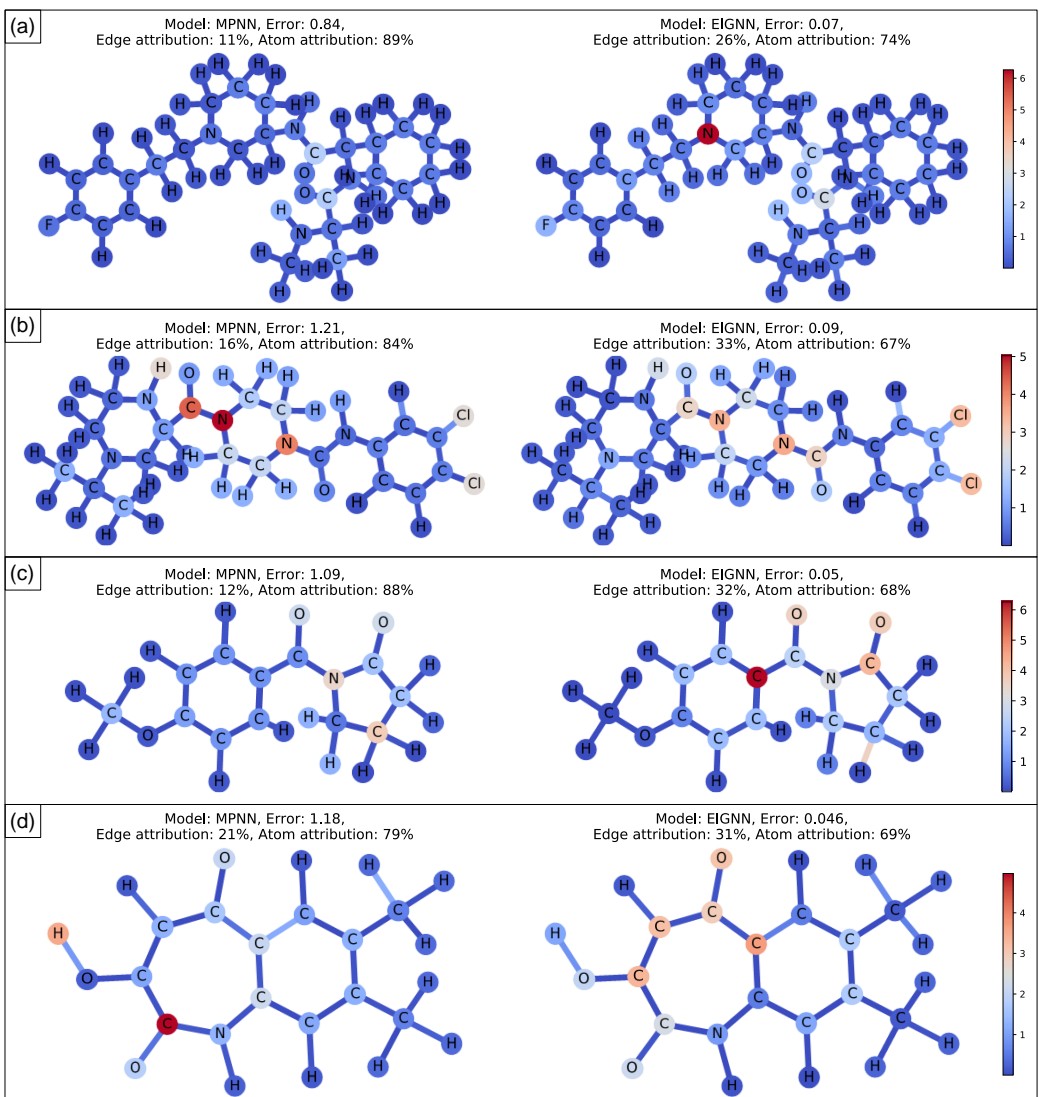

Figure 3: More examples of attribution analysis. The color indicates the impact of an edge/atom on the output, i.e., the regression result. EIGNN i) increases the edge attribution, ii) reduces the prediction error and iii) can learn domain knowledge without human interference.

