# OpenReview forum: "Utilizing Edge Features in Graph Neural Networks via Variational Information Maximization"
_ICLR.cc/2020/Conference — Reject_

### Official Review · AnonReviewer2 · 2019-10-21
**Official Blind Review #2**

**Rating:** 6

**Review:**

This paper proposed a new graph neural network to utilize the edge features. In particular, it proposes the Edge Information maximized Graph Neural Network (EIGNN) that maximizes the Mutual Information (MI) between edge features and message passing channels. The MI is reformulated as a differentiable objective via a variational approach. The experimental results have some improvement over existing methods.  Overall, the idea is novel and well presented.

Pros:
1. The idea of utilizing edge features looks novel.
2. The writing is clear.
3. Extensive experiments are done to verify the performance of the proposed method.

Cons:
1. The theoretical analysis is just a regular routine.
2. How does the hyper-parameter affect the performance? In other words, how does the component of the edge features affect the model performance?

**Experience Assessment:**

I have published in this field for several years.

**Review Assessment: Checking Correctness Of Derivations And Theory:**

I assessed the sensibility of the derivations and theory.

**Review Assessment: Checking Correctness Of Experiments:**

I assessed the sensibility of the experiments.

**Review Assessment: Thoroughness In Paper Reading:**

I read the paper thoroughly.

---

> ### Author Response · Authors · 2019-11-10
> **To AnonReviewer2. Thank you for your overall positive feedback.**
>
> 1. The theoretical analysis is just a regular routine.
> Yes, the theoretical analysis is based on well-established techniques. Experts can easily understand and verify our work. Nevertheless, the outcome (the proposed loss with MI being maximized) of this theoretical analysis is new, and its effectiveness in boosting performance is thoroughly studied and corroborated in experiments.
>
> 2. The component of the edge features.
> We surely agree with you that the component of edge features does affect the model performance, which actually supports our claim on the importance of edge features and our motivation of better utilizing the features. The influence of the component of edge features has been directly demonstrated in our experiments. For example, In Table 1, we have shown that MPNN can outperform sMPNN by using additional edge features, i.e., the pairwise distance between atoms.
>
> Edge feature design is a topic of feature engineering and requires domain knowledge according to specific applications. To demonstrate the effectiveness of our EIGNN, we follow previous literature and use the same feature engineering for fair comparison. Given the same available features, our method stands out better in terms of utilizing edge features.

---

### Official Review · AnonReviewer3 · 2019-10-22
**Official Blind Review #3**

**Rating:** 8

**Review:**

In this paper, the authors proposed a new kind of graph neural network that can use continuous edge features. Specifically, a variational lower bound is proposed for mutual information and integrated into the GNN model so that MI between edge features and the message passing channel is maximized. Experiments on molecule graph datasets and knowledge graph datasets show the effectiveness of the proposed method compared to the state-of-the-art GNN models.

Utilizing continuous edge features in GNN is an important and difficult task. The authors proposed an elegant solution. The paper is well written and extensive experiments on large scale datasets are compared with 9 competitors that are all outperformed by EIGNN on nearly all the cases.

To conclude, I think the paper will be a good addition to the conference.

**Experience Assessment:**

I have published one or two papers in this area.

**Review Assessment: Checking Correctness Of Derivations And Theory:**

I assessed the sensibility of the derivations and theory.

**Review Assessment: Checking Correctness Of Experiments:**

I assessed the sensibility of the experiments.

**Review Assessment: Thoroughness In Paper Reading:**

I read the paper at least twice and used my best judgement in assessing the paper.

---

> ### Author Response · Authors · 2019-11-10
> **To AnonReviewer3. Thank you for your positive feedback.**
>
> We appreciate your positive feedback and will keep polishing our work. Thank you very much.

---

### Official Review · AnonReviewer1 · 2019-10-23
**Official Blind Review #1**

**Rating:** 3

**Review:**

This paper introduces a mutual information term into the training objective of message passing graph neural networks.  The additional term favors the preservation on information in a mapping from an input edge feature vector e_{i,j} to a weight matrix f(e_{i,j}) used in computing messages across the edge from node i to node j.  A variational lower bound on the mutual information is used in training.  Impressive empirical results are given for chemical property prediction and relation prediction in knowledge graphs.  I have no real complaints other than I might recommend citing the original work on infomax:

"Self Organization in a Perceptual Network", Ralph Linsker, 1988.

Postscript:  I have been swayed by the complaints of reviewer 1 and reduced my score to weak reject.

**Experience Assessment:**

I have published one or two papers in this area.

**Review Assessment: Checking Correctness Of Derivations And Theory:**

I carefully checked the derivations and theory.

**Review Assessment: Checking Correctness Of Experiments:**

I assessed the sensibility of the experiments.

**Review Assessment: Thoroughness In Paper Reading:**

I read the paper at least twice and used my best judgement in assessing the paper.

---

> ### Author Response · Authors · 2019-11-10
> **To AnonReviewer1. Thank you for your positive feedback.**
>
> We appreciate your positive feedback and thank you for pointing out the original paper on infomax ([Linsker 1998]) which explains the importance of applying the infomax principle for designing neural networks. We have cited the paper and discussed it in Section 3.1.
>
> [Linsker 1998] Linsker, Ralph. "Self-organization in a perceptual network." Computer 21.3 (1988): 105-117.

---

### Official Review · AnonReviewer4 · 2019-10-26
**Official Blind Review #4**

**Rating:** 3

**Review:**

This paper proposed an auxiliary loss based on mutual information for graph neural network. Such loss is to maximize the mutual information between edge representation and corresponding edge feature in GNN ‘message passing’ function. By assuming a gaussian distribution of edge feature given edge representation, the training can be done efficiently with tractable density. Experiments on molecule regression and knowledge graph completion show better performance than MPNN.

Overall the paper is written in a clear way which is easy to follow. The idea of using mutual information as some kind of regularization is also interesting. However, there are some concerns I have with the paper:

Regarding formulation

1. The derivation up to Eq(8) looks fine to me, where the assumptions are reasonable. However from Eq(8) one can see this is reduced to an ‘auto-encoder’ type of regularization, where one can have a trivial solution for reconstruction -- the identity network, when the hidden dimension is larger than input dimension. And in this paper, dimension of W should always be larger than the dimension of e (for example, in molecules e should be low dimension vector with the bond type, distance, etc., while W should have dimension that matches the node embeddings).

I think the original loss (i.e., the supervised MSE, cross entropy etc) would help a bit with such degenerated case, but it is possible that the learned f(e) contains both identity mapping (or equivalent) and the representation that contributes to original loss.

2. Actually I’m also not sure if I get the motivation here. If one needs to do this regularization for edges, why don’t we consider this auxiliary loss for node embeddings as well? As in molecules, atoms have more interesting features than bonds, which should account more if the mutual information loss is needed.

Regarding experiment

1. In Figure 1, the training loss of EIGNN is better than MPNN. This is a bit counterintuitive to me, as I think the auxiliary is a kind of regularization -- which might help with generalization but not necessarily the training loss.

2. The original paper of MPNN reports the relative MAE. Is it possible to report the results using the same metric as previous paper? It would make the comparison more consistent -- though showing the improvement in current way is not too bad.

3. I think one simple ablation study would to concat the edge feature directly inside the ‘message passing’ procedure, or have some ‘residual’ type of connection for edge features.


**Experience Assessment:**

I have published in this field for several years.

**Review Assessment: Checking Correctness Of Derivations And Theory:**

I carefully checked the derivations and theory.

**Review Assessment: Checking Correctness Of Experiments:**

I assessed the sensibility of the experiments.

**Review Assessment: Thoroughness In Paper Reading:**

I read the paper thoroughly.

---

> ### Author Response · Authors · 2019-11-10
> **To AnonReviewer4. Thank you for your feedback and insightful discussions.**
>
> Your concerns are insightful. We updated the paper to add some discussions and experimental results and polish the presentation according to your feedback.
>
> Regarding formulation
>
> 1. A degenerated case of containing identity mapping.
> Thank you for pointing out this. We surely agree with your remarks on the possible degenerated case (i.e. f(e) containing identity mapping) but this is not a bad thing if it naturally emerges from a training process. If this emerges, what we obtain is a solution that maximizes the mutual information and minimizes the original loss as well. Then, this is actually a feasible optimal solution, which does not affect our main conclusion.
>
>
> 2. Motivation and why not applying to node embedding.
> Our work is motivated by the lack of focus on improving performance of existing Graph Neural Networks (GNNs) through better utilization of edge features. The idea is built upon well-established information theory, experts can easily judge its usefulness and/or limitations for specific applications. Simple ideas tend to be more universally applicable. This rule-of-thumb works well in our case: the MI loss can be applied to many GNN architectures without much coding efforts, and the solid performance boosts across a wide range of applications (from small to large graphs) chosen in our experiments surely motivated us to report this finding.
>
> We thank the referee for this insightful idea, and fully agree that applying our MI loss to node embedding is worth trying. However, the key emphasis of this work is how to better utilize edge features in GNNs. We think it is best to perform the suggested analysis in a future study.
>
>
>
> Regarding experiment
>
> 1. The training loss of EIGNN is better than MPNN.
> Our paper has been updated to discuss this phenomenon. In the updated figure 1, we show the standard deviation over 3 runs, which verifies that the results are statistically significant. Hence, this is an experimentally reproducible phenomenon. More discussions are provided in the main text in Section 4.1.
>
> Reduced training loss for EIGNN can be attributed to our key observation in this work: “the improvement of EIGNN over baselines is not due to simple regularization effect (like L_2 regularization) but comes from exploiting edge features.” This does seem to be counterintuitive if we take the auxiliary MI loss as a regularization. A possible explanation is that the auxiliary MI loss enforces better exploitation of edge features and smooths the optimization landscape, assisting the network to escape local optima. However, we admit that this is simply an intuitive explanation and topics on the optimization landscape and local optimums remain as a future work.
>
> 2. The original paper of MPNN reports the relative MAE [Gilmer et al. 2017].
> The reason we use MAE instead of relative MAE is that it is common to report MAE in terms of chemical unit. This practice has been widely adopted not only in computational chemistry but also in the machine learning community working on molecular graphs [Morris et al. 2019, Wu et al. 2018]. It can be found that our MPNN results are consistent with [Morris et al. 2019] (Table 2), [Wu et al. 2017] (Table 12) and implementations in the widely used libraries for graph-based deep learning such as [PyG] and [DGL].
>
> We would like to further emphasize that the relative MAE is simply the ratio of MAE to some predefined constants (i.e., the so-called chemical accuracy) and possibly involves unit change. For easy comparison, we added Table 5 in Appendix A which clearly lists all targets with descriptions and units. Nevertheless, according to your suggestion, we added Table 6 in Appendix A which contains the relative MAE for MPNN and our EIGNN. As pointed out in your comments, the improvement shown in our paper shall be consistent regardless whether it is reported in terms of relative MAE or original MAE.
>
> 3. One simple ablation study would to concat the edge feature directly inside the ‘message passing’ procedure, or have some ‘residual’ type of connection for edge features.
> According to your advice, we concatenate edge features to node representations in message passing and repeated the experiments 3 times. We updated Figure 1 in our paper to report the additional results. Concatenation is unable to identify correlations between edges and nodes (This is also claimed in Section 2 of [Gilmer et al. 2017]). Our results show that by concatenating edge features, the model achieves slightly better mean validation accuracy but the variance of accuracy is larger.
>
>
> Reference:
> [Gilmer et al. 2017] Neural Message Passing for Quantum Chemistry.
> [Morris et al. 2019] Weisfeiler and Leman Go Neural: Higher-Order Graph Neural Networks.
> [Wu et al. 2018] MoleculeNet: A Benchmark for Molecular Machine Learning. (https://arxiv.org/pdf/1703.00564.pdf)
> [PyG] PyTorch Geometric. (https://github.com/rusty1s/pytorch_geometric)
> [DGL] Deep Graph Library. (https://github.com/dmlc/dgl)

---

### Decision · Program_Chairs · 2019-12-19

**Decision:**

Reject

**Comment:**

This paper proposed an auxiliary loss based on mutual information for graph neural network. Such loss is to maximize the mutual information between edge representation and corresponding edge feature in GNN ‘message passing’ function. GNN with edge features have already been proposed in the literature. Furthermore,  the reviewers think the paper needs to improve further in terms of explain more clearly the motivation and rationale behind the method.